# Mycoviruses in the Rust Fungus *Uromyces fabae*

**Janina M. Seitz, Ralf T. Voegele**  **and Tobias I. Link ***

Department of Phytopathology, Institute of Phytomedicine, Faculty of Agricultural Sciences,
University of Hohenheim, Otto-Sander-Straße 5, 70599 Stuttgart, Germany; janina.m.seitz@gmail.com (J.M.S.);
ralf.voegele@uni-hohenheim.de (R.T.V.)
* Correspondence: tobias.link@uni-hohenheim.de; Tel.: +49-711-459-22392

**Abstract:** *Uromyces fabae*, the causal agent of broad bean rust, is a major cause of yield losses in North and East Africa, China, and Australia. It has also served as an important model species for research on rust fungi. Early EST sequencing in *U. fabae* showed that viruses might be present in this species; however, no follow-up investigations were conducted. In order to identify these viruses, we performed purification of dsRNA followed by Illumina sequencing. We also used ultracentrifugation followed by negative staining electron microscopy to visualize virus particles. We identified 20 viral sequences, which we termed Ufvss. A phylogenetic analysis was performed that grouped Ufvss into totiviruses, polymycoviruses, and virgavirus; three sequences could not be included in the phylogeny. We also found isometric particles. Our findings contribute to the knowledge of mycoviral diversity in rust fungi and point to the importance of further investigation of these viruses.

**Keywords:** *Uromyces fabae*; broad bean rust; mycovirus; dsRNA; virus-like particles; phylogeny

## 1. Introduction

*Uromyces fabae* (*U. vicia-fabae*) is the causal agent of broad bean rust or faba bean rust. It is a member of the Puccinales, the rust fungi. It is an autoecious and macrocyclic rust fungus [1]. As an obligate biotrophic pathogen, *U. fabae* depends on living host tissue for its nutritional needs and completing its life cycle [1]. Its host plant, *Vicia faba* (broad bean), is both an important crop species and a forage crop. *U. fabae* is one of the main diseases of *V. faba* in North and East Africa, China, and Australia. Yield losses of up to 68% have been reported in susceptible varieties [2]. Besides broad bean, *U. fabae* can cause rust disease in pea (*Pisum sativum* L.), lentil (*Lens culinaris* Medic.), and more than 50 other *Vicia* as well as 20 *Lathyrus* species [3]. For more than six decades *U. fabae* has been used as a model species to study rust infection structures and nutrient uptake [1].

Mycoviruses are fungal viruses that "infect and replicate in fungi" [4]. They are common in all major taxonomic groups of true fungi and frequent in plant-pathogenic oomycetes. The first mycovirus was described in the early 1960s on cultivated mushrooms (*Agaricus bisporus*) [5]. The existence of double-stranded ribonucleic acids (dsRNAs) in rust fungi was described for the first time by Newton et al. in 1985 [6]. They found dsRNAs in isolates of the cereal rusts *Puccinia striiformis*, *Puccinia recondita*, and *Puccinia hordei*. According to Zhang et al. [7] most rust fungi possess large amounts of dsRNAs; the quantity and size seem to be species-specific. Screenings showed that mycoviruses seem to be prevalent in 30–80% of phytopathogenic fungi [4]. Most of the mycoviruses cause cryptic infections, which means that they have no obvious effect on their fungal hosts. Some mycoviruses can lead to a decrease (hypovirulence) or an increase (hypervirulence) in fungal virulence [4]. Although research on mycoviruses has been going on for 60 years, the role of mycoviruses and the origin of mycoviruses are still largely unknown [4,8].

Two hypotheses concerning the origin of mycoviruses have been formulated: the "ancient coevolution hypothesis", which focuses on the long-term association and coevolution between mycoviruses and fungi and the "plant-virus hypothesis", which states that

mycoviruses were originally plant viruses that changed their hosts from plants to plant pathogenic fungi [9].

According to the International Committee on Taxonomy of Viruses (ICTV), a large number of mycoviruses have been discovered. These have been classified, so far, into 23 families and one additional genus that could not be assigned to a family [10]. Most typically, mycoviruses have a dsRNA genome and virus particles are isometric (*Totiviridae*, *Chrysoviridae* and *Partitiviridae*) [4]. Two-thirds of the mycoviruses possess a dsRNA genome, while one-third have a single-stranded RNA genome [11]. Mycoviruses with dsRNA genomes mostly have isometric particles between 30 and 80 nm in size. They can be non-segmented (*Totiviridae*) or have two (*Partitiviridae*, *Botybirnavirus*, *Megabirnaviridae*), four (*Chrysoviridae*, *Quadriviridae*), or ten to twelve genome segments (*Reoviridae*) [10].

The presence of dsRNAs of viral origin in *U. fabae* was first reported by Jakupovic et al. [12]. Through EST sequencing, these researchers found three ESTs with lengths between 675 and 940 bp that exhibited homologies to ssRNA plant viruses (cucumber fruit mottle mosaic virus and cucumber green mottle mosaic virus). As independent evidence for the putative presence of viruses in *U. fabae* and to show that this finding was not due to plant contamination of their RNA preparation (which was from haustoria isolated from *V. faba* leaves), they reported the presence of several dsRNA bands in a DNA preparation from spores with sizes ranging between 0.6 kb and 5.0 kb.

Here, we report the sequences of viruses of *U. fabae* obtained by sequencing dsRNA. These sequences are set in a phylogenetic context with other viral sequences downloaded as close relatives after BLAST searches. We also identified isometric virus particles.

## 2. Materials and Methods

### 2.1. Plant Material, Fungal Isolate and Spore Propagation

Broad bean (*Vicia faba* L.) cultivar Con Amore (Nickerson-Zwaan, Edemissen) was cultivated in the greenhouse under 16 h light and 8 h dark at 22 °C for three weeks. *Uromyces fabae* (PERS.) SCHROETER Isolate I2 (laboratory collection, Phytopathology, University of Hohenheim) was used for spore propagation. The inoculation suspension (0.01% Tween20, 0.2% milk powder, 0.2% *U. fabae* urediospores) was stirred for 45 min at room temperature. Broad bean plants were sprayed with the suspension using a chromatography sprayer (Carl Roth GmbH, Karlsruhe, Germany). The broad bean plants were then incubated for 16 h at 20 °C under 100% humidity in plastic boxes. The plants were returned from these boxes to the greenhouse under the above conditions. The first harvesting of urediospores was conducted 12–14 days post-inoculation (dpi).

### 2.2. Double-Stranded RNA Isolation from Urediospores

Isolation and purification of dsRNA were performed using the protocol by Morris and Dodds [13], with some modifications. This protocol makes use of the specific binding of dsRNA to cellulose in the presence of 16.5% ethanol.

A sample of 1 g *U. fabae* urediospores was used for the isolation of dsRNA. Urediospores were ground in liquid nitrogen and suspended in a mixture of 20 mL Sodium Chloride-Tris-EDTA (STE: 0.1 M NaCl, 0.05 M Tris, 1 mM Tris HCl, pH 6.85) buffer with 15 mL phenol:chloroform:isoamyl alcohol in the ratio 75:75:1 and 5 mL 10% SDS. This was stirred for 30 min at room temperature (RT) using a magnetic stirrer. For phase separation, centrifugation was conducted for 20 min at 9300 rcf and 4 °C.

The aqueous phase was spiked with ethanol up to a concentration of 16.5%. 1.5 g cellulose was added and the suspension was stirred for 30 min at RT. The suspension was then centrifuged for 3 min at 100 rcf, the supernatant discarded, and the pellet resuspended in 20 mL STE containing 16.5% ethanol. This was centrifuged and resuspended again until the supernatant was clear. The pellet of the final centrifugation was resuspended in 20 mL STE containing 16.5% ethanol and transferred onto a chromatography column (Poly-Prep® Bio-Rad Laboratories).

Before adding the above suspension, the column was washed twice with ddH$_2$O and twice with STE containing 16.5% ethanol. After adding the suspension, the column

was washed three times with 5 mL STE containing 16.5% ethanol. To remove all liquids, especially ethanol, air pressure was applied to the column with a manual air pump. dsRNA was eluted using 20 mL STE.

Three more steps were performed to increase the purity of the dsRNA. The first was a digest with RNase T1 (Thermo Fisher Scientific, Waltham, MA, USA). In order to ensure that the enzyme only digests ssRNA, it was diluted with ddH$_2$O to 1 u/μL and 20 μL of this was added to the above eluate. The mixture was incubated at 37 °C for 30 min. To stop the reaction, 1/30 vol 1 M MgCl$_2$ was added. Next was a digest with 40 μL (1 u/μL) DNase I (Thermo Fisher Scientific) for 30 min at 37 °C. This reaction was stopped by adding 1/15 vol 0.5 M EDTA.

The dsRNA was purified once more by adsorption chromatography. For this, 1 g of cellulose was added to the dsRNA solution, which was then brought to contain 16.5% ethanol. Another Poly-Prep® column was pretreated as described above and the suspension applied to it. The column was washed with STE containing 16.5% ethanol, dried as described above, and the dsRNA eluted using 3 mL STE.

For precipitation, 0.1 vol 3 M Na-Acetate, pH 5.2, and 2.5 vol ethanol were added. The dsRNA was stored in this state at −20 °C. For use, dsRNA was pelleted, washed with 70% ethanol, and solubilized in ddH$_2$O.

### 2.3. DNA Isolation from Urediospores

DNA isolation was performed according to the protocol by Kolmer et al. [14]. This was followed by an RNase T1 digest; so that, in effect, genomic DNA and dsRNA were prepared.

Briefly, 0.5 g of urediospores was ground in liquid nitrogen and acid-washed sand. The homogenate was added to 5 mL CTAB lysis buffer (2.5% *w/v* sorbitol, 1% *w/v* SDS, 0.8% *w/v* Cetyltrimethylammonium bromide (CTAB), 4.7% *w/v* NaCl, 1% *w/v* Na-EDTA, 1% *w/v* Polyvinylpyrrolidone (PVP), pH 8). 50 μL proteinase K (10 μg/μL) was added. The tube containing the mixture was shaken and then incubated at 58 °C for 1 h. 5 mL chloroform was added and the mixture centrifuged at 23,000 rcf for 15 min at 4 °C.

The aqueous phase was transferred to a new tube and 4 μL RNase T1 (1 u/μL) was added. The mixture was incubated at 37 °C for 30 min and then 5 mL chloroform was added. This was followed by centrifugation at 23,000 rcf for 5 min at 4 °C. Again, the aqueous phase was transferred to a new tube. For precipitation, 1/10 vol 3 M Na-Acetate (pH 5.2) and 4 mL cold isopropanol were added. This was followed by incubation at 4 °C for 30 min and centrifugation at 23,000 rcf for 20 min at 4 °C. The resulting pellet was washed with 2 mL 70% ethanol and then dried for 15 min at RT. Finally, DNA was solubilized in 25 μL ddH$_2$O.

### 2.4. Purification of Virus-like Particles and Electron Microscopy

The protocol by Sanderlin and Ghabrial [15], with modifications, was used for the purification of virus-like particles. A 2 g sample of *U. fabae* urediospores was ground in liquid nitrogen and suspended in 7 mL 50 mM potassium phosphate buffer (pH 7) and 0.1% (*w/v*) Na$_2$SO$_3$. This mixture was centrifuged for 15 min at 10,000 rcf and 8 °C. The supernatant of the first centrifugation was centrifuged at 75,000 rcf and 8 °C for 3 h to pellet the particles. This supernatant was discarded and the pellet solubilized overnight at 4 °C in 500 μL Tris-HCl (pH 7.5). To remove the remaining debris, this solution was again centrifuged at 5000 rcf for 5 min and the pellet discarded.

Negative staining was performed using a protocol by Mulisch [16], with modifications. A copper grid (400 mesh with carbon coating) was put onto a 20 μL droplet of the particle solution for 10 min to bind the particles. The grid was then left to dry. For negative staining, the grid was positioned on a 20 μL droplet of 2% (*w/v*) uranyl acetate (pH 4.3) for 10 min. Again, the grid was left to dry after removing excess liquid using filter paper.

Transmission electron microscopy was conducted using an EM 10 (Zeiss, Oberkochen, Germany) and the negatives of the pictures were scanned.

*2.5. Sequencing and Assembly*

Illumina sequencing, sequence assembly, and identification of virus sequences were conducted by VirSeq Services at the Plant Virus Department of the Leibniz-Institut, DSMZ— Deutsche Sammlung von Mikroorganismen und Zellkulturen GmbH (Hannover, Germany). The dsRNA was sent there precipitated in ethanol (see Section 2.2).

Library preparation from dsRNA was performed using the Nextera XT kit (Illumina, San Diego, CA, USA). The library was sequenced using Illumina MiSeq using a paired-end reads protocol (2 × 301). This generated 3,098,536 reads that were assembled *de novo*. The first 1000 contigs (based on number of reads per contig) were BLASTed against the plant and fungal databases in NCBI.

Contigs identified as viral were trimmed and combined with other contigs.

*2.6. Sequence Analysis*

For a first classification, standard BLASTn was performed against the virus database in NCBI. Open reading frames (ORFs) were predicted using GeneRunner (version 6.5.52 beta) and checked by a person. Alignments for phylogenetic analysis were performed using ClustalW as implemented in BioEdit [17]. The phylogenetic analysis itself was performed using MEGAX (version 10.0.5) [18]. The maximum likelihood method was run using the following settings: Bootstrap method (100 bootstrap replications), amino acid Le_Gascuel_2008 (LG) matrix-based model [19], and rates among sites (uniform rates). Tree inference options: ML heuristic method: nearest neighbor interchange (NN). Initial Tree for ML by Neighbor-Join and BioNJ, make initial tree automatically, branch swap filter very strong.

## 3. Results

*3.1. Seven Fractions of Double-Stranded RNA Can Be Distinguished in U. fabae*

Due to existing knowledge from other fungi and mycoviruses, and also because dsRNA had already been observed in *U. fabae*, we based our search for mycoviruses in *U. fabae* on establishing the presence of dsRNA. To do this, dsRNA was prepared from *U. fabae* urediospores using a protocol based on the specific binding of dsRNA to cellulose in 16.5% ethanol. When the dsRNA was separated on a 1% agarose gel, seven distinct bands were observed (Figure 1a).

To corroborate this finding, an alternative protocol was used. This was a protocol for DNA preparation followed by RNase T1 digestion. This procedure results in a DNA preparation that may also contain dsRNA. When the DNA was separated on a 1% agarose gel, dsRNA bands similar to those observed in the dsRNA preparation were observed in addition to the genomic DNA (Figure 1b).

These results show that *U. fabae* contains dsRNA of at least seven different sizes, which indicates that it is infected with mycoviruses. Potentially, different dsRNA fragments can represent the genomes of different viruses or parts of the genome of one virus.

*3.2. Viruses of U. fabae Have Isometric Particles*

Since mycoviruses are only transmitted vertically or via anastomosis and have no extracellular phase [4], they can exist without particles to protect their nucleic acids. Nevertheless, particles have been described for some mycoviruses [4] and the existence of particles can be regarded as further evidence for infection with viruses. Therefore, we searched for virus particles in *U. fabae*. Since no particles have been observed in electron micrographs of *U. fabae* in the past (Kurt Mendgen, personal communication), we decided against sectioning infected plant material with *U. fabae* infection and tried particle purification instead. Since nothing was known about possible particles, we chose a general purification protocol based on ultracentrifugation (see Section 2.4).

Isometric particles were identified in the preparation from *U. fabae* urediospores (Figure 2). They were approximately 80–90 nm in diameter. This means that at least one of the viruses infecting *U. fabae* produces a coat.

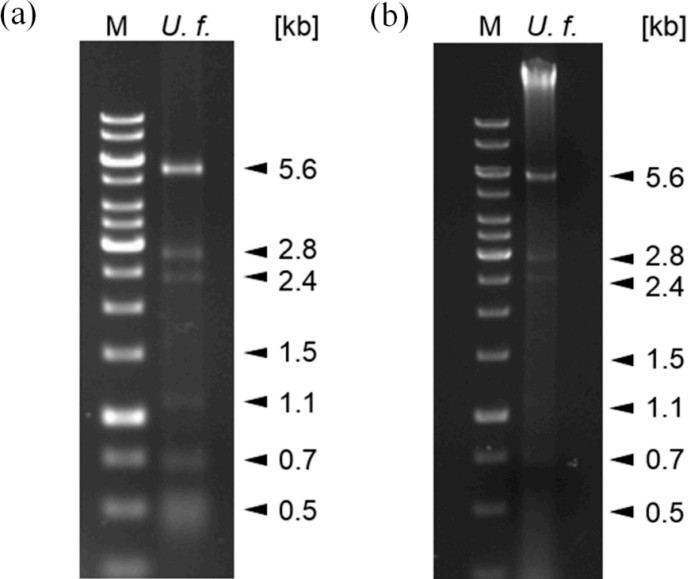

**Figure 1.** Distinct dsRNA fractions in nucleic acid preparations from *U. fabae*. (**a**) dsRNA: Agarose gel showing bands of dsRNA of putative viral origin from *U. fabae*. Roughly 10% (in 10 μL) of the dsRNA prepared as described in Section 2.2. (*U. f.*) and 5 μL GeneRuler 1 kb Plus DNA ladder (M) were electrophoretically separated on a 1% agarose gel run at 100 V for 60 min. (**b**) DNA and dsRNA: Agarose gel showing genomic DNA (strong band at the top of the lane) and bands of dsRNA of putative viral origin from *U. fabae*. Roughly 40% (in 10 μL) of the DNA prepared as described in Section 2.3. (*U. f.*) and 5 μL GeneRuler 1 kb Plus DNA ladder (M) were electrophoretically separated on a 1% agarose gel run at 100 V for 60 min.

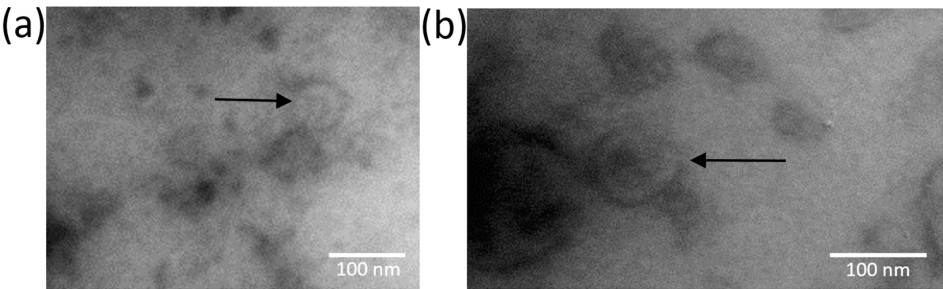

**Figure 2.** Electron micrographs of *U. fabae* virus particles. (**a**,**b**) sections from different pictures taken as described in Section 2.4. depicting putative isometric virus particles (arrows).

*3.3. Sequencing of dsRNA Yields Evidence for Several Viral Genomes*

dsRNA prepared as described in Section 2.2. was sequenced using Illumina MiSeq technology. This yielded 20 contiguous sequences with similarities to viral sequences. The sequences were named Ufvs_1–Ufvs_20 (Ufvs: *Uromyces fabae* viral sequence, ordered by size) and submitted to NCBI. Open reading frames were predicted from these sequences and BLASTp similarity searches were performed for a first classification of the viruses (Table 1).

RT-PCR followed by Sanger sequencing and production of a plasmid library from double-stranded cDNA was also performed. The resulting sequences and the sequences obtained by Jakupovic et al. [12] were compared to the above sequences in an assembly of assemblies. Most of the Sanger sequences did assemble to the Ufvss. A few sequences did not assemble, which indicates that more viruses may be present in *U. fabae*; however, these sequences were too short to yield reliable BLAST hits and, therefore, viral identity could not be confirmed. For Ufvs_12, the Sanger sequences that assembled to the contig also yielded additional information.

**Table 1.** Viral sequences in *U. fabae*.

| Virus | Accession | Length (nt) | Open Reading Frames | | | | | |
|---|---|---|---|---|---|---|---|---|
| | | | Position | Frame | AA | Top BLAST Hit | E Value | %ID |
| Ufvs_1 | OQ995224 | 10755 | 33–5366 | +3 | 1777 | methyltransferase, partial (Wheat-associated tobamo-like virus) UIN24825.1 | 0.0 | 64.36 |
| | | | 6959–9529 | +2 | 856 | DEAD-like helicase (Wheat-associated tobamo-like virus) UIN24854.1 | 0.0 | 60.56 |
| | | | 5589–6935 | +3 | 448 | RNA-dependent RNA polymerase (Wheat-associated tobamo-like virus) UIN24853.1 | 0.0 | 90.75 |
| | | | 9618–10511 | +3 | 297 | hypothetical protein (Wheat-associated tobamo-like virus) UIN24855.1 | $4 \times 10^{-164}$ | 77.74 |
| Ufvs_2 | OQ995225 | 5133 | 136–2607 | +1 | 823 | putative CP (Uromyces totivirus A) QED43025.1 | 0.0 | 70.81 |
| | | | 2691–5108 | +3 | 805 | RdRp, partial (Uromyces totivirus A) QED43026.1 | 0.0 | 71.58 |
| Ufvs_3 | OQ995226 | 5001 | 63–2519 | +3 | 818 | capsid protein (Helianthus annuus leaf-associated totivirus 6) UMQ74227.1 | 0.0 | 70.29 |
| | | | 3068–4981 | +2 | 637 | putative RdRp (Uromyces totivirus B) QED42952.1 | 0.0 | 70.65 |
| Ufvs_4 | OQ995227 | 4983 | 49–2532 | +1 | 827 | coat protein (Erysiphales-associated totivirus 6) QIP68055.1 | 0.0 | 43.70 |
| | | | 2721–4979 | +3 | 752 | RNA-dependent RNA polymerase (Erysiphales-associated totivirus 6) QIP68054.1 | 0.0 | 51.07 |
| Ufvs_5 | OQ995228 | 4971 | 24–2222 | +3 | 732 | putative capsid protein (Poaceae Liege totivirus 8) UVG55934.1 | $2 \times 10^{-175}$ | 39.53 |
| | | | 2222–4951 | +2 | 909 | putative RNA-dependent RNA polymerase (Poaceae Liege totivirus 8) UVG55935.1 | 0.0 | 47.87 |
| Ufvs_6 | OQ995229 | 4967 | 76–2481 | +1 | 801 | putative capsid protein (*Puccinia striiformis* totivirus 4) ATO91012.1 | 0.0 | 72.03 |
| | | | 2532–4922 | +3 | 796 | putative RNA-dependent RNA polymerase (*Puccinia striiformis* totivirus 4) ATO91013.1 | 0.0 | 67.93 |
| Ufvs_7 | OQ995230 | 4967 | 47–2521 | +2 | 824 | putative capsid protein (*Puccinia striiformis* totivirus 1) ATO91006.1 | 0.0 | 39.35 |
| | | | 2560–4941 | +1 | 793 | putative RdRp (Phakopsora totivirus B) QED42972.1 | 0.0 | 68.70 |
| Ufvs_8 | OQ995231 | 4778 | 25–2268 | +1 | 747 | capsid protein (Helianthus annus leaf-associated totivirus 7) UMQ74228.1 | 0.0 | 88.95 |
| | | | 2271–4751 | +3 | 826 | hypothetical protein 2 (Wuhan insect virus 26) YP_009342428.1 | 0.0 | 46.20 |
| Ufvs_9 | OQ995232 | 4758 | 276–2204 | +3 | 642 | hypothetical protein (Uromyces totivirus D) QED43018.1 | 0.0 | 80.53 |
| | | | 2243–4735 | +2 | 830 | hypothetical protein, partial (Uromyces totivirus D) QED43019.1 | 0.0 | 80.29 |
| Ufvs_10 | OQ995233 | 4708 | 84–2144 | +3 | 686 | putative capsid protein (Puccinia striiformis totivirus 5) ATO91014.1 | 0.0 | 80.47 |
| | | | 2327–4693 | +2 | 788 | putative RNA-dependent RNA polymerase (Puccinia striiformis totivirus 5) ATO91015.1 | 0.0 | 72.88 |
| Ufvs_11 | OQ995234 | 3054 | 89–2053 | +2 | 654 | putative capsid protein (*Puccinia striiformis* totivirus 1) ATO91006.1 | $2 \times 10^{-131}$ | 35.19 |
| | | | 2059–3012 | +1 | 317 | putative RdRp (Phakopsora totivirus E) QED42935.1 | $4 \times 10^{-75}$ | 41.70 |
| Ufvs_12 | OQ995235 | 2885 | 196–2847 | +1 | 883 | RNA-dependent RNA polymerase (Erysiphales associated totivirus 19) QIP68078.1 | 0.0 | 46.36 |
| Ufvs_13 | OQ995236 | 2430 | 34–1881 | +1 | 615 | putative RdRp (Phakopsora totivirus E) QED42935.1 | 0.0 | 55.90 |
| | | | 1826–2392 | +2 | 188 | RNA-dependent RNA polymerase (Red clover powdery mildew-associated totivirus 5) YP_009182188.1 | $7 \times 10^{-13}$ | 26.37 |
| Ufvs_14 | OQ995237 | 2130 | 31–2055 | +1 | 674 | RdRp, partial (Uromyces virus B) QED43024.1 | $2 \times 10^{-171}$ | 43.35 |

**Table 1.** *Cont.*

| Virus | Accession | Length (nt) | Open Reading Frames | | | | | |
| | | | Position | Frame | AA | Top BLAST Hit | E Value | %ID |
|---|---|---|---|---|---|---|---|---|
| Ufvs_15 | OQ995238 | 2097 | 211–2049 | +1 | 612 | RdRp, partial (Uromyces virus B) QED43024.1 | 0.0 | 94.92 |
| Ufvs_16 | OQ995239 | 2007 | 141–1925 | +3 | 594 | RNA-dependent RNA polymerase, partial (Helianthus annuus leaf-associated totivirus 3) UMQ74221.1 | 0.0 | 75.38 |
| Ufvs_17 | OQ995240 | 1592 | 27–1577 | +3 | 516 | putative RNA-dependent RNA polymerase (*Puccinia striiformis* totivirus 5) ATO91015.1 | 0.0 | 81.57 |
| Ufvs_18 | OQ995241 | 1560 | 45–1313 | +3 | 422 | p46 (Citrus concave gum-associated virus) UDN65939.1 | $4 \times 10^{-70}$ | 36.56 |
| Ufvs_19 | OQ995242 | 1123 | 994–212 | −1 | 260 | nucleocapsid (Laurel Lake virus) YP_009667030.1 | $2 \times 10^{-39}$ | 35.51 |
| Ufvs_20 | OQ995243 | 1113 | beg–753 | +1 | 250 | Movement protein (blackberry line pattern virus) WDD63193.1 | $5 \times 10^{-31}$ | 36.41 |

Using parameters with low stringency also generated a contig that contained both Ufvs_10 and Ufvs_17, and another contig that contained Ufvs_11 and Ufvs_13. This indicates that these viruses have highly similar genomes; however, close inspection of the assemblies also revealed that the sequences are different, thus we treated these Ufvss as distinct viruses.

Most of the sequences did not have the typical terminal sequences of mycoviruses. Together with the results of the assembly of assemblies, this indicates that the sequences are not yet complete. To ensure full genome sequences, it will be necessary to perform RACE PCR to obtain the terminal sequences of the viral genomes. It will also be necessary to perform PCR using outward-facing primers designed for the different sequences to amplify fragments between the sequences to possibly combine more of the fragments. It is conceivable that the smaller sequences (Ufvs_13–Ufvs_20) are part of a larger genome. It might be that some of them should be combined into one sequence or that there is a virus with a partite genome.

Nine of the Ufvss have lengths of approximately 5 kb. These could correspond to the 5.6 kb band on the gel (Figure 1). There is one Ufvs with 2.8 kb in length and two with 1.5 kb in length. These may correspond to the respective bands on the gel (Figure 1). The smaller bands on the gel might be fragments. Northern blots correlating the bands with the sequences have not yielded clear results so far.

### 3.4. Phylogenetic Analysis of the Viral Sequences

To obtain a better understanding of the viral diversity in *U. fabae* and a better classification of the viruses into families, the viral sequences were grouped phylogenetically. An additional aim of phylogenetic grouping was to gain insights into when and how *U. fabae* might have acquired the different viruses.

The phylogenies were based on predicted RdRp protein sequences. Since Ufvs_18, Ufvs_19, and Ufvs_20 did not yield an ORF for an RdRp, they were not included in the analysis.

In order to make the phylogeny informative, the RdRp protein sequences were also BLASTed and the best BLAST hits (based on %identity) together with additional BLAST hits of virus species that gave information on the systematic grouping were downloaded. All sequences were aligned and trees calculated based on maximum likelihood. The first tree that was built showed three obvious clusters: a very large cluster with totivirus sequences and two smaller groups with tobamo-like viruses and polymycoviruses. This phylogeny is not shown here due to its low resolution; however, it was used to separate the sequences falling into the three mentioned groups and to remove a few species, which did not prove as informative as assumed during BLASTing, from the analysis.

This resulted in three smaller assemblies containing the totiviruses, the tobamo-like or virgaviruses, and the polymycoviruses, respectively, and to the phylogenies shown in Figures 3–5. In all phylogenies, we have labeled our novel sequences in bold, except for the two sequences representing the other families that were included as outgroups. Where the closest relatives to our viral sequences are also from *Uromyces*, this is indicated in brackets labeled with "Uromyces virus". To shorten the identifiers of the taxa in the figures, information like the accession numbers of the sequences was removed. This information is provided in Table 2.

**Table 2.** RdRp sequences used for phylogenetic analyses (provided in alphabetical order).

| Virus Name | NCBI Accession No. of RdRp |
|---|---|
| Acidomyces richmondensis tobamo-like virus 1 | AZT88673:1 |
| Aspergillus fumigatus polymycovirus 1 | BCH36613.1 |
| Aspergillus spelaeus tetramycovirus 1 | YP_010839683.1 |
| Auricularia heimuer mycovirgavirus 1 | QIM57886.1 |
| Beauveria bassiana polymycovirus 1 | VCV25414.1 |
| Delisea pulchra totivirus IndA | AMB17468 |
| Delisea pulchra totivirus IndA | AMB17473 |
| Delisea pulchra totivirus IndA | AMB17477 |
| Delisea pulchra totivirus IndA | AMB17478 |
| Delisea pulchra totivirus IndA | AMB17470 |
| Diplodia seriata polymycovirus 1 | UOK20165.1 |
| Erysiphales associated totivirus 3 | QIP68048.1 |
| Erysiphales associated totivirus 4 | QIP68050.1 |
| Erysiphales associated totivirus 6 | QIP68054.1 |
| Erysiphe necator associated tobamo-like virus 1 | QKN22701.1 |
| Hubei toti-like virus 2 | YP_009336496.1 |
| Hubei toti-like virus 3 | APG76078.1 |
| Hubei toti-like virus 4 | APG76044.1 |
| Hubei virga-like virus 23 | YP_009337439.1 |
| Macrophomina phaseolina tobamo-like virus 1a-A | QOE55599.1 |
| Maize associated totivirus 1 | AWD38954.1 |
| Maize-associated totivirus 2 | YP_009259486.1 |
| Penicillium brevicompactum tetramycovirus 1 | YP_010086053.1 |
| Phakopsora pachyrhizi mycovirus | ALO81041.1 |
| Phakopsora totivirus A | QED42974.1 |
| Phakopsora totivirus C | QED43023.1 |
| Poaceae Liege totivirus 9 | UVG55937.1 |
| Poaceae Liege totivirus 10 | UVG55939.1 |
| Puccinia striiformis totivirus 1 | ATO91007.1 |
| Puccinia striiformis totivirus 2 | ATO91009.1 |
| Puccinia striiformis totivirus 3 | ATO91011.1 |
| Puccinia striiformis totivirus 4 | ATO91013.1 |
| Red clover powdery mildew-associated totivirus 1 | BAT62478.1 |
| Red clover powdery mildew-associated totivirus 2 | YP_009182176.1 |
| Red clover powdery mildew-associated totivirus 3 | YP_009182181.1 |
| Red clover powdery mildew-associated totivirus 4 | BAT62484.1 |
| Red clover powdery mildew-associated totivirus 6 | YP_009182190.1 |
| Red clover powdery mildew-associated totivirus 7 | YP_009182195.1 |
| Red clover powdery mildew-associated totivirus 8 | BAT62492.1 |
| Saccharomyces paradoxus virus L-A-45 | ATL63182.1 |
| Scheffersomyces segobiensis virus L | YP_009507831.1 |
| Sclerotinia sclerotiorum tetramycovirus-1 | AWY10945.1 |
| Trichoderma koningiopsis totivirus 1 | QGA70771.1 |
| Tuber aestivum virus 1 | YP_009507833.1 |
| Wuhan insect virus 26 | YP_009342428.1 |
| Wuhan insect virus 27 | YP_009342434.1 |
| Xanthophyllomyces dendrorhous virus L1A | YP_007697651.1 |
| Xanthophyllomyces dendrorhous virus L1B | YP_009507835.1 |
| XiangYun toti-like virus 8 | UUG74262.1 |

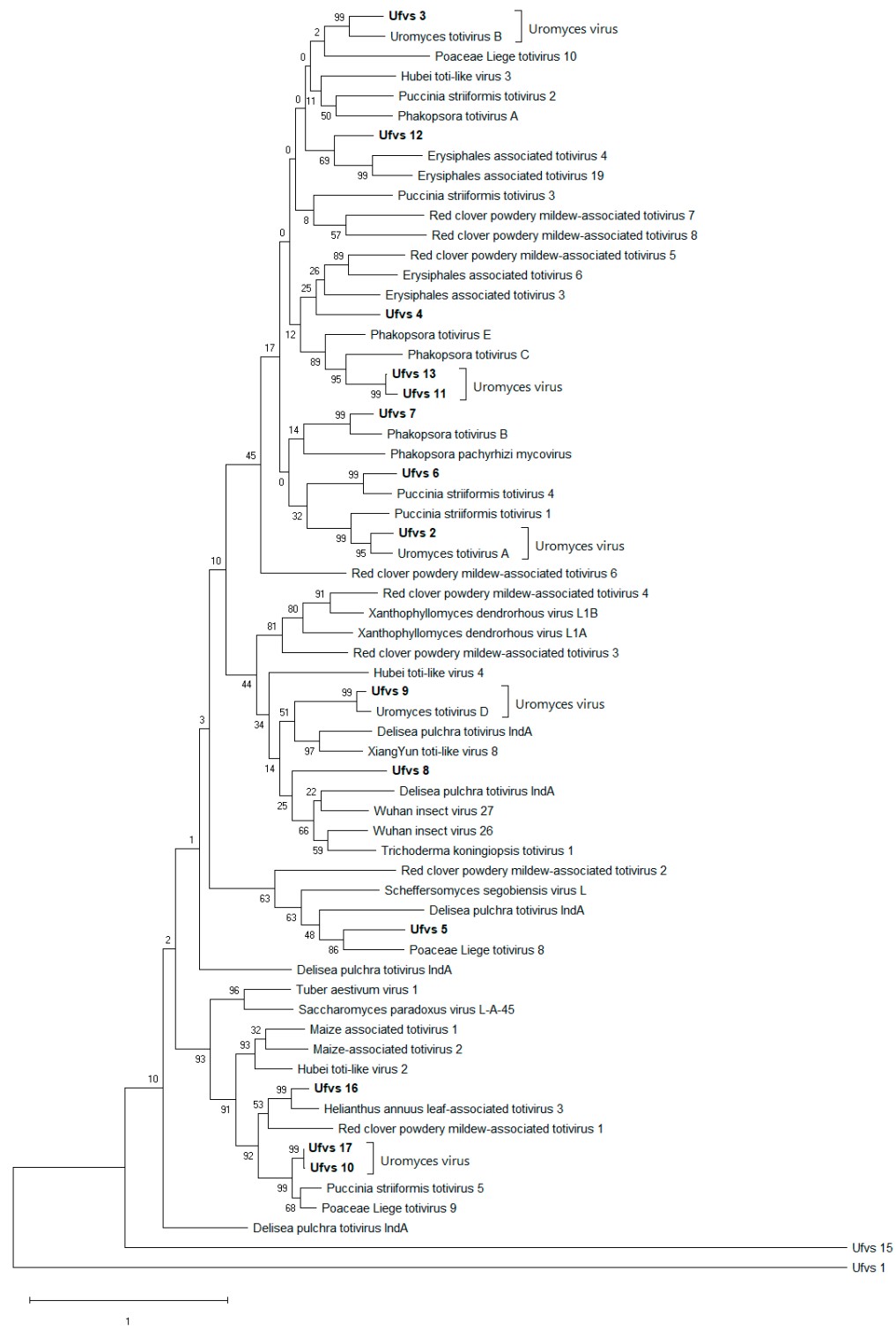

**Figure 3.** Totiviruses of *U. fabae* in relation to their close relatives–evolutionary analysis using maximum likelihood method. The tree with the highest log likelihood (-33,273.58) is shown. The percentage of trees in which the associated taxa clustered together is shown next to the branches. The tree is drawn to scale, with branch lengths measured in the number of substitutions per site. The analysis involved 64 RdRp sequences. All positions with less than 80% site coverage were eliminated, i.e., fewer than 20% alignment gaps, missing data, and ambiguous bases were allowed at any position (partial deletion option). There were a total of 367 positions in the final dataset. Ufvs_1 and Ufvs_15 were included as outgroups to represent tobamo-like viruses and polymycoviruses, respectively.

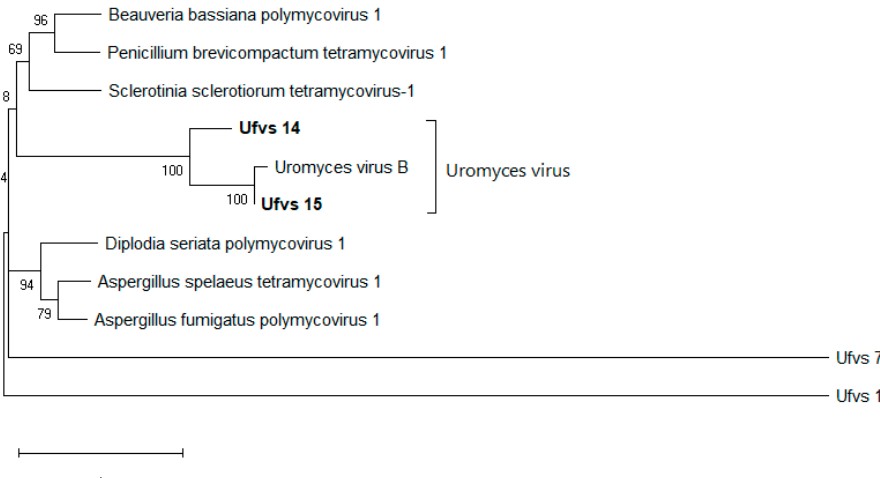

**Figure 4.** Polymycoviruses of *U. fabae* (Ufvs_14, Ufvs_15) in relation to their close relatives–evolutionary analysis using maximum likelihood method. The tree with the highest log likelihood (-9260.88) is shown. The percentage of trees in which the associated taxa clustered together is shown next to the branches. The tree is drawn to scale, with branch lengths measured in the number of substitutions per site. The analysis involved 11 RdRp sequences. All positions with less than 60% site coverage were eliminated, i.e., fewer than 40% alignment gaps, missing data, and ambiguous bases were allowed at any position (partial deletion option). There were a total of 482 positions in the final dataset. Ufvs_7 and Ufvs_1 were included as outgroups, representing totiviruses and tobamo-like viruses, respectively.

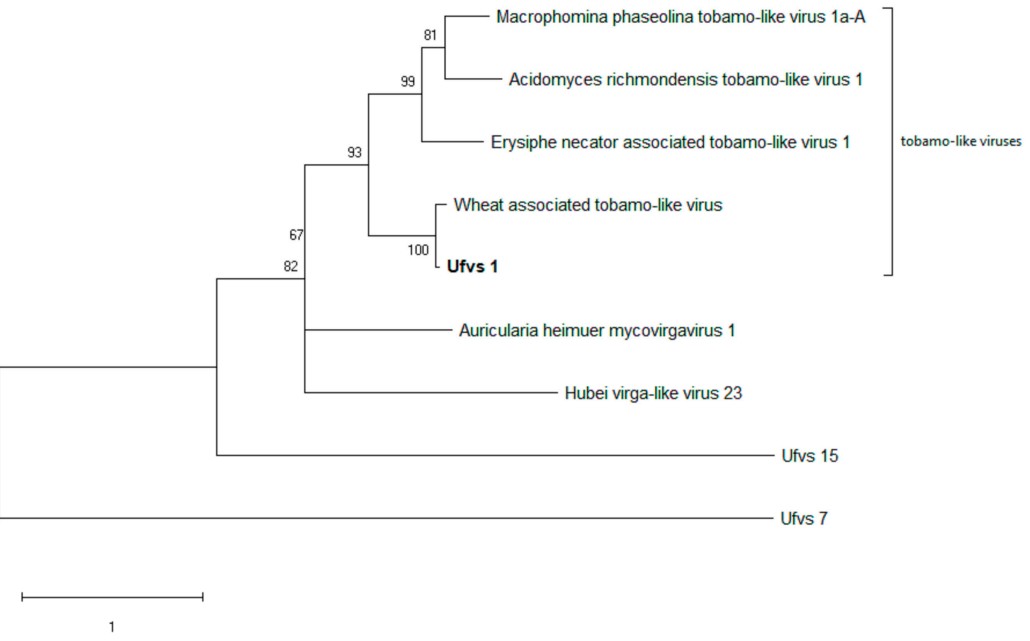

**Figure 5.** Ufvs_1 and its close relatives–evolutionary analysis using maximum likelihood method. The tree with the highest log likelihood (-8123.81) is shown. The percentage of trees in which the associated taxa clustered together is shown next to the branches. The tree is drawn to scale, with branch lengths measured in the number of substitutions per site. This analysis involved 9 RdRp sequences. All positions with less than 60% site coverage were eliminated, i.e., fewer than 40% alignment gaps, missing data, and ambiguous bases were allowed at any position (partial deletion option). There were a total of 449 positions in the final dataset. Ufvs_7 and Ufvs_15 were included as outgroups, representing totiviruses and polymycoviruses, respectively.

The phylogeny in Figure 3 indicates that most of the totiviruses have evolved from a common ancestor in the rust fungi. Apart from Ufvs_11 and Ufvs_13, and Ufvs_10 and Ufvs_17, where the closest relative is another virus in *U. fabae*, most closest relatives were found either within the genus *Uromyces*—as for Ufvs_2, Ufvs_3, and Ufvs_9 (the Uromyces viruses were found in *Uromyces appendiculatus*)—or in other species of the order *Pucciniales*—as for Ufvs_7, Ufvs_11, and Ufvs_13 (*Phakopsora pachyrhizi*), or Ufvs_6, Ufvs_10, and Ufvs_17 (*Puccinia striiformis*). Only Ufvs_4, Ufvs_5, Ufvs_8, Ufvs_12, and Ufvs_16 have their closest relatives outside the rust fungi, which might indicate that these viruses made a jump across taxa. With the current amount of data, however, no real conclusions can be drawn; it is quite likely that the closest relatives inside the rust fungi have not been sequenced, yet. In general, it must be said that while the totivirus family is well defined and the clusters inside the family are well supported, the branches defining the relationships between those clusters are unclear.

As can be seen in Figure 4, the closest relative of Ufvs_14 and Ufvs_15 was found in *U. appendiculatus*. The other species were included in the phylogeny since they indicate which virus family these viruses belong to. Apart from Uromyces virus B, these are polymycoviruses or tetramycoviruses. However, the similarity is not very high. The three viruses from *Uromyces* form a separate cluster in the phylogeny, set apart from but somehow in between two other clusters. The genus *Polymycovirus* is a new genus whose characterization has only just begun [20]. Generally, the polymycoviruses have at least four genome segments. Here, we were able to sequence only one segment, indicating either that there is still more to be found or that the other segments are absent and that the virus segment depends on some of the other viruses.

The viruses showing homology to Ufvs_1 all belong to the family *Virgaviridae*. These are generally rod-shaped plant viruses. The closest relative is a virus associated with wheat. While the authors reporting this virus did not experimentally resolve this, it is likely that the host of this virus is a fungus that infects wheat [21]. The next close relatives are mycoviruses, two of which are from plant pathogens. As the hosts of these viruses are all ascomycetes, there may have been a host jump.

## 4. Discussion

Here we present the identification of 20 virus-like sequences from *U. fabae*, based on the sequencing of dsRNA. We also found virus particles in *U. fabae* urediospores using ultracentrifugation and negative staining in transmission electron microscopy.

Recently, other groups have performed similar sequencing projects [22] or searched data from RNAseq projects [23] and found whole populations of viruses in the fungal isolates that were studied. This includes the rust species *U. appendiculatus*, *P. pachyrhizi* [24,25], and *P. striiformis* [26]. Likewise, sequencing in powdery mildews yielded several viral sequences [27]. This yields additional information on how these viruses are related to each other and to viruses other than mycoviruses.

Phylogenetic analysis grouped the viruses in *U. fabae* into three groups: totiviruses, polymycoviruses, and virgaviruses. All three groups of viruses are virus families containing mycoviruses or that are mainly composed of mycoviruses [21,26,27]. The findings here were as expected. Three putative viruses, which could not be grouped phylogenetically, were found; these may belong to other families.

A persistent enigma of mycoviruses is how they evolve, especially for viruses in plant pathogenic fungi this is hard to resolve. They could have coevolved with their hosts, or they could have been taken up by their hosts from plants or other fungi [9]. This is interesting not only because of the question of how easy host jumps can occur with these viruses but also because mycoviruses are almost completely intracellular. Free particles and new infections have not been observed [10,28].

The phylogeny of the totiviruses among the Ufvss provides some evidence that these mycoviruses had a long evolution in fungi rather than in plants since no closely related plant viruses were identified. How the viruses evolved within the kingdom fungi is a bit

more uncertain. The phylogeny shows instances where the species clearly evolved inside their host and only recently separated, like Ufvs_10 and Ufvs_17 or Ufvs_11 and Ufvs_13, or the group of Red clover powdery mildew-associated totiviruses 6, 7, and 8, which cluster together. There are three instances (Ufvs_3 and Uromyces totivirus B, Ufvs_2 and Uromyces totivirus A, and Ufvs_9 and Uromyces totivirus D) where the closest relative of the virus in *U. fabae* is a virus of the closely related rust fungus *U. appendiculatus*, which also suggests that these viruses evolved with the hosts—the virus species may have separated together with the host species. The same was observed for the polymycoviruses Ufvs_15 and Ufvs_14, which have a close relative in Uromyces virus B, also from *U. appendiculatus*. There are more close relatives in *P. pachyrhizi* and *P. striiformis*, which also indicates that the viruses evolved over a long time inside the *Pucciniales*.

The interpretation of Ufvs_4 and Ufvs_12 clustering with totiviruses from powdery mildews is not so straightforward. On the one hand, the related viruses also infect plant pathogenic fungi, hence they have probably evolved as mycoviruses. On the other hand, *Erysiphales*, the powdery mildews, are ascomycetes and phylogenetically far removed from *Pucciniales*, which are basidiomycetes. In these instances, it seems more likely that when there was a common infection of a plant—likely a legume—the virus made the jump between the two plant pathogens [27].

Other clustering gives less information. For the maize-associated totiviruses, Helianthus annuus leaf-associated totivirus 3, or the virgaviruses, wheat-associated tobamo-like virus, it is difficult to decide whether these are plant viruses or viruses of a plant pathogenic fungus infecting maize, sunflower, or wheat. The same is true for the potentially insect-associated Hubei viruses [29].

Altogether, our BLAST and phylogenetic data seem to indicate that these viruses evolved in fungi, although it is not entirely clear whether host jumps between fungi were involved.

Our finding of virus particles—but possibly only one kind of particles—is in accordance with findings for other mycoviruses, which do not have particles as they exist only cytoplasmically, whereas others do produce particles [4]. To which of the Ufvss the particles belong is impossible to tell at this stage, since we identified 10 ORFs coding for (putative) coat proteins. To investigate this, it will be necessary to purify the particles further using gradient centrifugation, which can potentially separate different particles by size. RNA preparation from these particles and RT-PCR can then be used to identify the Ufvs corresponding to the particles.

The large number of different virus species infecting *U. fabae*, or rather our isolate I2, which was propagated in the lab for quite some time, seems astonishing—at least 20 different viruses infecting the same organism at the same time! Given that mycoviruses do not actively infect fungi but are transmitted only vertically or through anastomosis, it is surprising that the viruses persist.

Part of the reason why this research was initiated was the thinking that viruses could be used for biological control of *U. fabae* or other rust fungi. This is done with *Cryphonectria parasitica* using Cryphonectria hypovirus 1 (CHV1) [30,31], and there was the thinking that this could also be established for rust fungi. Since our isolate performed well on *V. faba* and we do not yet have a virus-free isolate to compare it with, at the moment it seems unlikely that one of the viruses that we found causes disease in *U. fabae*, or that we can find it.

On the other hand, since they do not actively infect their hosts, to be able to persist, mycoviruses should not confer fitness costs to their hosts but instead should offer a fitness benefit, otherwise they would be randomly lost and die out [9]. Since they are using the resources of their hosts, viruses should theoretically always come with some fitness cost and to compensate for that, they would need to confer some benefit. With 20 or more viruses infecting *U. fabae*, it seems quite likely that one or more of them confers fitness benefits.

It would now be highly interesting to identify these fitness benefits. With a plant pathogen, it is always interesting to look for virulence factors. Since the genomes of the viruses are largely uncharacterized, it is certainly possible that these genomes contain

virulence factors for the rust fungus that make them indispensable for the pathogen. Indirect evidence for such virulence factors has been found for two Puccinia striiformis viruses [32,33]. If these virulence factors could be identified, it would be feasible to eliminate them by infecting the fungus with a virus mutant that does not contain it; some of the virulence factors might even define host specificity. Our *U. fabae* isolate infects *V. faba*; however, there is still controversy about whether the species should still be called *Uromyces viciae-fabae* since it is also an important pathogen in pea and lentil and can infect more than 50 other *Vicia* species and roughly 20 *Lathyrus* species [1]. There is also some evidence that this is a species complex with different species or subspecies that still have to be defined infecting different species [34]. In this context, it could be interesting to compare our *U. fabae* isolate with *U. viciae-fabae* isolates infecting pea for their virus complements.

Finding these virulence factors will (or would) be a major research project. Not only would it need virus-free isolates of the rust fungus that can be re-infected with the different virus species to see which virus confers which virulence factor, it would also require cloning all virus sequences into infectious clones to enable the infections, and finally systematically deleting parts of the virus genomes in these infectious clones to find the actual virulence factor.

## 5. Conclusions

We have identified several virus sequences in *U. fabae*. These sequences contribute to the knowledge of mycovirus diversity in general and the rust fungi in particular. It is also important to know that these viruses are present in the fungus. Given that mycoviruses are notoriously understudied, the simple awareness of the fact that these viruses probably influence the biology of the fungi they infect is a bonus. Investigating these viruses further to identify this influence will be a major effort.

**Author Contributions:** Conceptualization, J.M.S., R.T.V. and T.I.L.; methodology, J.M.S. and T.I.L.; formal analysis, J.M.S. and T.I.L.; investigation, J.M.S.; data curation, T.I.L.; writing—original draft preparation, J.M.S. and T.I.L.; writing—review and editing, J.M.S., R.T.V. and T.I.L.; visualization, J.M.S. and T.I.L.; project administration, T.I.L.; funding acquisition, J.M.S. All authors have read and agreed to the published version of the manuscript.

**Funding:** This research was funded through a scholarship from the Faculty of Agricultural Sciences of the University of Hohenheim to Janina M. Seitz. The APC was funded by the University of Hohenheim.

**Institutional Review Board Statement:** Not applicable.

**Informed Consent Statement:** Not applicable.

**Data Availability Statement:** All relevant data are contained within this publication. Sequences were submitted to NCBI, accession numbers are provided.

**Acknowledgments:** We are grateful to Annerose Heller and Barbara Kaufmann for their expertise and assistance with virus particle purification and electron microscopy. Artur Pfitzner provided ideas for the initial planning of this research.

**Conflicts of Interest:** The authors declare no conflict of interest.

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
