# Peer review of "Mycoviruses in the Rust Fungus Uromyces fabae"

_viruses, doi:10.3390/v15081692_

Round 1

Reviewer 1 Report

The paper no. viruses-2519097 on the characterization of mycoviruses in the rust fungus Uromyces fabae. The authors performed purification of dsRNA followed by Illumina sequencing to identity mycoviruses in U. fabae. Furtherly, the authors used ultracentrifugation followed by negative staining electron microscopy to visualize virus particles. The results of phylogenetic analysis showed that mycoviruses were grouped into totiviruses, polymycoviruses, virgaviruses, and three additional sequences, which could not be treated phylogenetically. These findings contribute to the knowledge on mycoviral diversity in rust fungi and point to the importance of further investigation of these viruses.

Comments and Suggestions for Authors

1. The name of mycoviruses U. fabae should be objectivized, for example, Uromyces fabae totivirus, but not Ufvs1, Ufvs2, etc.

2.  The isometric viral-like particles from U. fabae could not be identified from which kind of virus. The detection result of mycoviruses in the isometric particles preparation by RT-PCR should be provided.

3. The results of sequences of identity should be provide in Table 1.

4. The quality of Figure 3-5 should be improve.

Reviewer 2 Report

Dear authors,

I read your paper concerning identifying mycoviruses in the rust fungus Uromyces fabae. Throughout the drafting of the manuscript, the novelty and innovation in reporting a descriptive analysis is evident. However, overall, the manuscript lacks the right form of communication. English needs to be implemented and some formulas, such as discussion questions, must be removed. I think the work can take advantage of the more technical and sectoral mode of expression rather than a form of questionnaire addressed to the reader. Then, I report some points that should be implemented.

1)     Line 23, Rust fungi doesn’t include only Pucciniales; correct and report update references.

2)     Moderate editing of English language required from abstract to conclusion. Check all the manuscripts.

3)     Liter in capital letter, e.g. microliter (µL).

4)     At least 50% of the references should be included articles published in the last 5 years; check. Read and cite:

-        Villegas-Fernández ÁM, Amarna AA, Moral J, Rubiales D. Crop Diversification to Control Rust in Faba Bean Caused by Uromyces viciae-fabae. J Fungi (Basel). 2023 Mar 11;9(3):344. doi: 10.3390/jof9030344. PMID: 36983512; PMCID: PMC10057490.

-        Barilli E, Rubiales D. Identification and Characterization of Resistance to Rust in Lentil and Its Wild Relatives. Plants (Basel). 2023 Jan 31;12(3):626. doi: 10.3390/plants12030626. PMID: 36771710; PMCID: PMC9919313.

-        Fei W, Liu Y. Biotrophic Fungal Pathogens: a Critical Overview. Appl Biochem Biotechnol. 2023 Jan;195(1):1-16. doi: 10.1007/s12010-022-04087-0. Epub 2022 Aug 11. PMID: 35951248.

5)     A translation approach should be included. For example, Mycoviruses can impact virulence. You should report previous descriptions in this field related to Pucciniales

Moderate editing of English language required

Round 2

Reviewer 2 Report

Dear Authors,

all the corrections have been made and the manuscript has been improved. 
